# The Diagnostic Capacity of Physical Examinations in Diagnosing Musculoskeletal Disorders of the Lower Limbs in Children with Down Syndrome

**DOI:** 10.3390/medicina59111986

**Published:** 2023-11-10

**Authors:** Barbara Lima Machado, Ronny Rodrigues Correia, Gabriela Alencar Pereira, Ieda Hiromi Maemura, Catia Regina Branco Fonseca, Pedro Luiz Toledo de Arruda Lourenção

**Affiliations:** 1Department of Surgery and Orthopedics, Botucatu Medical School, São Paulo State University (UNESP), Botucatu 18618-687, Brazil; bahmachado@hotmail.com (B.L.M.); ronny.rodrigues@unesp.br (R.R.C.); 2Department of Pediatrics, Pediatric Gastroenterology, Hepatology and Nutrition, Botucatu Medical School, São Paulo State University (UNESP), Botucatu 18618-687, Brazil; gabriela.alencar@unesp.br (G.A.P.); iedahiromi95@gmail.com (I.H.M.); catia.fonseca@unesp.br (C.R.B.F.)

**Keywords:** down syndrome, children, musculoskeletal disorders, lower limbs

## Abstract

*Background and Objectives*: although musculoskeletal alterations are common in patients with Down syndrome (DS), studies investigating this association are scarce, and proposals for diagnostic standardization are limited. We aimed to evaluate the prevalence of musculoskeletal disorders in the lower limbs in a sample of children and adolescents with DS and to investigate the diagnostic capacity of orthopedic clinical examinations performed by orthopedists and pediatricians to diagnose these alterations. *Materials and Methods*: Twenty-two patients aged between three and ten years with DS were included. Patients and guardians answered a simple questionnaire regarding orthopedic complaints and underwent a systematic orthopedic physical examination, performed twice: once by an orthopedist and again by a pediatrician. Patients underwent a series of radiographs to diagnose anisomelia, hip dysplasia, epiphysiolysis, flatfoot valgus, mechanical axis varus, and mechanical axis valgus. The radiological diagnosis was considered the gold standard, and the diagnostic capacity of the physical examination performed by each physician was determined. *Results*: The median age was 6.50 years. Only four patients (18.2%) presented with orthopedic complaints. All patients were diagnosed with at least one musculoskeletal disorder. The only musculoskeletal disorder with a good diagnostic capacity was flatfoot valgus. Limited sensitivity values were found for hip dysplasia, mechanical axis varus, and mechanical axis valgus. The agreement between the orthopedic physical examinations performed by the two examiners was weak, poor, or indeterminate for most of the analyzed items. *Conclusions*: There was a high prevalence of orthopedic alterations in children with DS who did not present with musculoskeletal complaints. The diagnostic capacity of the physical examination was limited. Therefore, all children with DS should undergo a radiological evaluation of the musculoskeletal system and subsequent specialized orthopedic evaluation. Level of Evidence: Level II (Diagnostic Studies).

## 1. Introduction

Down syndrome (DS) is the most common chromosomal alteration in humans and is the leading cause of intellectual disability in the population. DS is a genetic anomaly characterized by an extra chromosome at position 21 in 92 to 95% of patients. In addition, DS may also be the result of mosaicism in 2 to 4% of patients or translocations in 3 to 4% of cases [1]. According to data from the Brazilian Ministry of Health, a child is born with DS in every 600 to 800 births, regardless of ethnicity, sex, or social class [2].

Children and adolescents with DS present with cardiovascular, endocrine, respiratory, and musculoskeletal disorders. Among the musculoskeletal disorders common in children with DS, atlantooccipital instability, flat foot valgus, patellar subluxation, femoropatellar instability, joint hypermobility, hip dysplasia, and developmental delays in the locomotive system stand out [1,3,4]. Although patients with DS also present alterations in the upper limbs, such as ligament laxity and hypotonia, alterations in the lower limbs are considered more relevant because they involve more significant energy expenditure and are directly related to postural adjustments and motor delays, such as sitting down, standing, and moving around [5].

Although musculoskeletal alterations are common in patients with DS and include many etiologies, studies investigating this association are scarce [1,6]. Many studies have found that DS is related to complications of various systems, such as the cardiovascular, dental, and endocrinological systems, but there are few reports on musculoskeletal complications [1,7]. In addition, proposals for diagnostic standardization and systematic referral for specialized orthopedic evaluations are limited [1,6].

Thus, the present study aimed to evaluate the prevalence of musculoskeletal disorders in the lower limbs in a sample of children and adolescents with DS and to investigate the diagnostic capacity of orthopedic clinical examinations performed by orthopedists and pediatricians to diagnose these alterations.

## 2. Material and Methods

This cross-sectional clinical study was developed at the genetic pediatrics outpatient clinic at the Botucatu Medical School Hospital, São Paulo State University (UNESP), Brazil, between January and December 2020. This study focused on the prevalence of musculoskeletal alterations in children with DS and on the diagnostic accuracy of orthopedic clinical examinations performed by orthopedists and pediatricians to diagnose these musculoskeletal alterations. First, patients and guardians responded to a questionnaire about possible orthopedic complaints. The patients then underwent systematic orthopedic physical examinations performed separately by an orthopedist and a pediatrician. Finally, a series of X-rays were taken to diagnose possible musculoskeletal disorders. The data obtained were analyzed to assess the prevalence of musculoskeletal disorders in the sample, the diagnostic capacity of the clinical examination, and the agreement between the results of the clinical examinations carried out by the orthopedist and the pediatrician. The local Research Ethics Committee approved the study under protocol CAAE no.29200620.8.0000.5411.

### 2.1. Eligibility Criteria and Sample Size

The inclusion criteria were patients of both sexes, aged between three and ten years, with DS confirmed via karyotype, with previous clinical and radiological evaluation measurements of the atlantoaxial index ruling out a diagnosis of atlantoaxial instability. The exclusion criteria were patients with regular orthopedic follow-ups or a lack of consent from their guardians.

A sample size of 22 patients was established. This sample size is based on an expected sensitivity of 90% for diagnosing musculoskeletal disorders of the lower limbs following clinical examination, considering a prevalence of 70% for musculoskeletal alterations in patients with DS, with a 15% margin of error.

### 2.2. Clinical Evaluations

Patients and guardians answered a simple questionnaire regarding orthopedic complaints applied by the same member of the research team responsible for asking the questions and recording the answers (Table 1). During the second follow-up appointment, patients underwent a systematic orthopedic physical examination, which included gait and physical inspections of the hip, knee, and feet; these evaluations were performed per a protocol developed specifically for the study (Table 2). The systematized orthopedic physical examination was performed twice: once by an orthopedist (examiner 1) and again by a pediatrician (examiner 2). Both physicians had the same number of years of experience and underwent qualification and training sessions before performing the systematized orthopedic examinations. Each physician was shielded from the results obtained in the assessment performed by the other physician.

Through systematic physical examination, each physician diagnosed the following alterations: anisomelia, hip dysplasia, epiphysiolysis, flatfoot valgus, mechanical axis varus, and mechanical axis valgus. The criteria for the diagnosis of anisomelia included the identification of asymmetry in the measurement of the lower limbs. Lower limb asymmetry was identified through measurements made with a measuring tape; the anterior superior iliac spine and medial malleolus of the same limb were used as measuring limits. A positive Galeazzi test was also used to denote asymmetry in the lower limbs [8]. The criteria used for clinical diagnosis of hip dysplasia included a positive Trendelenburg test or identification of hip adduction contracture and consequent abduction limitation [8,9]. The criteria for clinical diagnosis of epiphysiolysis included gait difficulty, hip abduction, and limitations to internal rotation [10]. The diagnosis of flatfoot valgus was based on identifying at least one of the following criteria: hindfoot valgus, a positive tiptoe test, a positive Jack test, and a positive too-many-toes test [11]. With the patient in an orthostatic position, the alignment of the lower limbs was subjectively found to be in the neutral, varus, or valgus positions [12]. The criteria for patella instability diagnosis included accurate identification of the “J” sign, a positive Fairbank apprehension test, and positive patellar tilt tests [13].

### 2.3. Radiological Evaluations

We proposed radiographs of the hip (anteroposterior, false lateral, and double abduction views (Lauenstein), knees (anteroposterior and lateral views), and a panoramic view of the lower limbs and feet (anteroposterior and lateral views with load views). Some patients could not have all of the proposed radiographs carried out due to technical difficulties (children’s collaboration and acceptance). A senior orthopedist with extensive experience in the field analyzed the radiographs.

Hilgenreiner and Perkins’ lines were traced on hip radiographs in the anteroposterior view to identify hip dysplasia. Hilgenreiner’s line was obtained by connecting the bilateral triradiate cartilages or from horizontal lines passing through the inferior surface of the iliac bones. Perkins’ vertical line was defined as a line perpendicular to Hilgenreiner’s line that crossed the lateral acetabular rim. These lines form Ombredanne Quadrants. The femoral head was located in the inferior and medial quadrants. The Wiberg center–lateral edge angle was calculated between the line passing through the center of the femoral head and perpendicular to the transverse axis of the pelvis, the line passing through the center of the head, and the superolateral point of the acetabular roof. Values greater than 25° were considered altered. Altered values indicated inadequate femoral head coverage and instability [8]. The acetabular index was measured between the Hilgenreiner line and the acetabular socket. Values greater than 25° were considered altered [8]. The diagnosis of hip dysplasia was determined in patients who presented with the femoral head not located in the inferior and medial Quadrants of Ombredanne, a Wiberg angle outside normal limits, or alterations in the acetabular index [8].

Anisomelia or dysmetria was evaluated using panoramic radiography [14]. From this image, the anatomical axes of the femur and tibia of each limb were traced; they corresponded with the longitudinal axis of the diaphysis. The sum of the measurements of the long axes of the femur and tibia represented the final measurement of the anatomical axis. The final measurement was compared with the contralateral side to determine the presence of dysmetria. The panoramic radiograph was also used to evaluate the lower limbs for the presence of varus or valgus along the mechanical axis. The mechanical axis of the entire lower limb is formed by a straight line connecting the center of the femoral head to the center of the ankle. A varus deformity was diagnosed when the line passed medially to the center of the knee. A valgus deformity was diagnosed when the line passed laterally to the center of the knee [8,14].

The lateral radiological evaluation of the knees was performed with the knee flexed at 30° to assess the patellar height. The two indices used to assess the lateral radiograph were the Caton–Deschamps index and the Blackburne–Peel index. The Caton–Deschamps index measures the relationship between the distance from the inferior pole of the articular surface of the patella to the anterosuperior border of the tibia and the length of the articular surface of the patella. Values less than 1.2 on the Caton–Deschamps index diagnosed a high patella. The Blackburne–Peel index measures the relationship between the size of the perpendicular line tangent to the tibial plateau at the inferior pole of the articular surface of the patella and the articular surface of the patella. Values above 1.0 on the Blackburne–Peel index diagnosed a patella alta [15].

Radiographs for visualization of the valgus flatfoot were taken in both anteroposterior and lateral views with a load. The Kite, Calcaneal Pitch, and Meary angles were evaluated in this study. Kite’s angle was formed between the long axes of the talus and the calcaneus. Values between 20° and 40° in the anteroposterior (AP) view and 35° and 50° in the lateral view were considered normal. Values exceeding the limits of normality in both incidences allowed for the diagnosis of flatfoot valgus. The calcaneal Pitch angle was determined using the plantar edge of the calcaneus and the horizontal surface only in the lateral view. Values between 15° and 25° were considered normal. Values lower than 15° allowed the diagnosis of valgus flatfoot. Meary’s angle was determined by the axis between the first metatarsal and the talus; this angle does not present with angulations in normal feet [8]. A diagnosis of valgus flatfoot was made in patients who presented with at least one of the angles mentioned above outside of the normal limits [16].

Diagnostic confirmation of epiphysiolisthesis was performed using simple radiographs in an anteroposterior view of the pelvis and the “frog” position or double abduction (Lauenstein) position. In the Lauenstein position, prior to displacement of the epiphysis with the neck, it is possible to evaluate the height or increased thickness of the growth plate. The growth plate may also become smooth, “bald”, and lose its characteristic serrated appearance in epiphysiolisthesis [10].

### 2.4. Diagnosis of Musculoskeletal Disorders of the Lower Limbs

Radiological diagnosis was considered the gold standard and was used to establish the diagnosis of anisomelia, hip dysplasia, epiphysiolysis, flatfoot valgus, mechanical axis varus, and mechanical axis valgus (Figure 1). Patellofemoral instability was diagnosed using clinical criteria. These clinical criteria consisted of the “J” sign, the Fairbank apprehension test, the patellar tilt test, and a radiological criterion; this radiological criterion evaluated the primary criterion for the recurrence of instability, which was the high patella. The Caton–Deschamps and Blackburne–Peel indices were measured on the lateral radiographs, and measurements outside of the normal limits of these indices denoted a high patella.

All patients diagnosed with musculoskeletal disorders were referred for treatment and clinical follow-up at our institution’s specialized orthopedic outpatient clinic.

### 2.5. Statistical Analysis

The diagnostic capacity of systematic physical examinations, performed by an orthopedist (examiner 1) and a pediatrician (examiner 2), was evaluated using radiological studies as the gold-standard tests to diagnose anisomelia, hip dysplasia, epiphysiolysis, valgus flat foot, varus mechanical axis, and valgus mechanical axis. This analysis cannot be performed for patellofemoral instability. Patellofemoral instability was diagnosed based on clinical and radiological criteria.

The sensitivity, specificity, positive predictive value, negative predictive value, accuracy, and area under the receiver operating characteristic curve (ROC curve) of the physical examination performed by each physician were determined. In addition, Cohen’s kappa statistics were used to analyze the findings of the orthopedic studies conducted by both examiners. Cohen’s kappa statistics, therefore, determined the kappa value, 95% confidence interval, and statistical significance [17,18].

Continuous numerical data were expressed as medians (minimum/maximum), according to the type of non-parametric distribution of the data previously determined according to the Shapiro–Wilk normality test. Proportions were presented as percentages with their respective 95% confidence intervals. In addition, continuous numerical variables with non-parametric distributions were compared using the Mann–Whitney U test. The significant level was set to 5%. Analyses were performed using SPSS 22.0 for Windows software.

## 3. Results

### 3.1. Clinical and Demographic Data

Forty-five patients of both sexes, with an age range of three-to-ten years, with a diagnosis of DS (confirmed by karyotype) were included. Nine patients undergoing regular orthopedic follow-ups and fourteen children whose guardians did not consent to their participation in the study were excluded. Thus, 22 patients (11 (50%) males and 11 (50%) females) were ultimately included in the study. The median age of the patients was 6.50 years; the youngest patient was 4 years old, while the oldest patient was 10 years old. Only four patients (18.2%) reported orthopedic complaints in their responses to the questionnaire. There was no significant difference between the median age of the group of patients with or without orthopedic complaints at the time of assessment (8.5 (4/10) versus 6.0 (4/10); *p* = 0.48; Mann–Whitney U test).

### 3.2. Prevalence of Lower Limb Musculoskeletal Disorders

All 22 patients underwent at least one of the proposed radiological examinations. Radiological assessment was used to diagnose one of the musculoskeletal alterations investigated in this study (Table 3). All patients were diagnosed with at least one musculoskeletal disorder. The distribution of the number of musculoskeletal disorders per patient is shown in Table 4.

### 3.3. The Diagnostic Capacity of Physical Examinations

Table 5 presents the indicators of the diagnostic capacity of physical examinations performed by examiners 1 and 2. The only musculoskeletal disorder for which there was a good diagnostic capacity (area under the curve (AUC) > 0.80 was flatfoot valgus. Reasonable diagnostic capabilities (AUC > 0.70) were found for the physical examination of anisomelia performed by examiner 2 and for the mechanical axis valgus performed by examiner 1.

### 3.4. Physical Examination Agreements between the Examiners

The concordance indicators of the physical examinations performed by examiners 1 and 2 for each item of the systematized orthopedic physical examination protocol are presented in Table 6. Concordance was considered weak, poor, or indeterminate for most items.

## 4. Discussion

All DS children in the study had at least one musculoskeletal disorder diagnosed via a clinical and radiological evaluation. On the other hand, less than 20% had related clinical complaints, highlighting the need for systematic orthopedic assessments, regardless of the symptoms. Orthopedic alterations in children with DS may cause functional impairments. These functional impairments are responsible for the increased energy expenditure required for DS children to remain upright and perform essential activities [16]. Integrating patients with DS into society, with reasonable cognitive control, depends on adjustments in gross and fine movement and body stability. Structural stability, matched by the musculoskeletal system, is crucial for establishing connections with others and generating social, economic, and personal well-being [1,6]. However, regular follow-up of such patients by orthopedists commonly does not follow protocols and depends on random referrals to referenced services [1,6,7].

Musculoskeletal diagnoses in our study had higher occurrence rates than in the literature. Previous reports described that approximately 20% of all patients with Down syndrome experience orthopedic problems [19,20]. Patellofemoral instability had a prevalence of 65% in our study; however, a previous study reported a prevalence of 8.3% of this disorder in 210 institutionalized patients with DS and of 4.0% in 151 non-institutionalized patients with Down syndrome patients with DS [21]. The prevalence of hip dysplasia in our study was 4.8%; this agreed with the 1.25 to 7% prevalence found in the literature [22]. We identified the following prevalences: (1) 82.3% for anisomelia; (2) 95.4% for flat foot valgus; (3) 52.9% for mechanical axis valgus; and (4) 29.4% for mechanical axis varus. Perotti et al. (2018) found a flatfoot prevalence of 58% in DS children less than 10 years of age, 59% in DS children between 10 and 13.9 years of age and 57% in DS children > 14 years of age [23]. In our study, a single patient had epiphysiolysis. Children with endocrinopathies, mainly hypothyroidism, are at high risk for epiphysiolysis of the proximal femur [10]. In our sample, all patients had regular follow-ups to monitor thyroid dysfunction at the childcare clinic. This may account for the low prevalence of epiphysiolysis observed in our study. The only patient diagnosed with epiphysiolysis was four years old, had congenital hypothyroidism, and had been unable to walk since birth.

Our results demonstrate the good diagnostic capacity of physical examinations to diagnose flatfoot valgus, regardless of the examiner’s specialty (AUC > 0.8). The limited sensitivity values for hip dysplasia, mechanical axis varus, and mechanical axis valgus highlight the limitations of physical examinations as a diagnostic screening tool for these musculoskeletal disorders of the lower limbs in children with Down syndrome. Furthermore, the agreement between the orthopedic physical examinations performed by the two examiners was weak, poor, or indeterminate for most of the analyzed items. These discrepancies demonstrate the considerable variation between physical examinations performed by different examiners, decreasing the reliability of physical examinations [24,25]. It must also be noted that this limited agreement occurred even with a systematized protocol used by trained examiners.

There are important limitations and difficulties in performing orthopedic physical examinations in children with DS. On average, each orthopedic examination took 10 min, with several repetitions of the same topic to permit the children to understand and accept the examinations and special tests. Agitation, crying, non-acceptance, and running away frequently impaired the accuracy of measurements and specific tests. It is difficult to transmit neuronal impulses in patients with DS. They generally have fewer dendritic branches and decreased communication between the different brain areas. Clinically, such findings may account for the children’s cognitive abilities and attention deficits while performing associated commands during physical examinations [26]. In addition, DS patients have auditory and attention deficits that make maintaining permanent focus throughout the proposed activities difficult [27].

The present study has some limitations that should be highlighted. First, it was a single-center study, which limited the sample size and generated bias. An example of this bias is demonstrated by the fact that all patients in the study had undergone regular follow-ups for thyroid function. This regularity in endocrinological follow-up may have influenced the prevalence of patients with epiphysiolysis. Other limitations were that some patients had not undergone all of the proposed radiological examinations and that the clinical examinations were carried out by only two examiners with little experience and different specialties, which may have influenced the assessments of diagnostic capacity and agreement.

On the other hand, our study presents some notable strengths. This study assessed the prevalence of primary musculoskeletal disorders of the lower limbs in children with DS. The study also evaluated the diagnostic capacity of physical examinations performed by orthopedists and pediatricians. The results of this study can broaden the debate on this topic and be used to propose changes in clinical practice.

Based on the results of this study, we can conclude that given the diagnostic limitations of orthopedic clinical examinations, radiographic examinations are the gold standard for diagnosing musculoskeletal alterations in the lower limbs of children with DS. These radiologic examinations should be routinely performed, regardless of the results of physical examinations. Based on the high prevalence of orthopedic alterations found in a sample of children with DS who did not present with musculoskeletal complaints and the limitations presented by the physical examination by orthopedists and pediatricians, we believe that all children with DS should undergo radiological evaluation of the musculoskeletal system and subsequent specialized orthopedic evaluation. These evaluations will permit early diagnoses of musculoskeletal alterations and prevent possible future debilitating complications. This assessment should preferably be carried out after three years of age since patellar ossification begins in this age group [28].

## Figures and Tables

**Figure 1 medicina-59-01986-f001:**
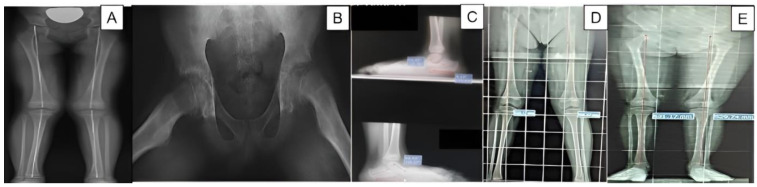
Radiographs of patients in the study. (**A**) lower limb dysmetria; (**B**) epiphysiolisthesis; (**C**) valgus flat foot; (**D**) valgus mechanical axis; (**E**) right varus mechanical axis.

**Table 1 medicina-59-01986-t001:** Directed clinical questionnaire on orthopedic complaints.

	Yes	No
- Does the patient have pain in the lower limbs at rest?		
- Does the patient have pain in the lower limbs when walking?		
- Does the patient have a gait imbalance?		
- Is there an inability to walk short and medium distances?		

**Table 2 medicina-59-01986-t002:** Systematized orthopedic physical examination protocol.

	Physical Examination
Gait Evaluation	Claudication Yes ( ) No ( ) Limb asymmetry (Galeazzi test) Yes ( ) No ( ) Trendelenburg sign Yes ( ) No ( ) Lower limb asymmetry determined from measurement Yes ( ) No ( )
Hip	Limitation on abduction Yes ( ) No ( ) Flexion > 130° Yes ( ) No ( ) Extension > 30° Yes ( ) No ( ) Thomas test Yes ( ) No ( ) Internal rotation > 45° Yes ( ) No ( ) External rotation > 50° Yes ( ) No ( ) Abduction > 45° Yes ( ) No ( ) Adduction > 40° Yes ( ) No ( ) Trendelenburg test Yes ( ) No ( )
Knee	Varus axis Yes ( ) No ( ) Valgus axis Yes ( ) No ( ) Q-angle increase Yes ( ) No ( ) Position of the patella with the knee flexed: in front of the femoral condyles Yes ( ) No ( ) Positive patellar tilt test (for evaluation of retinacula) Yes ( ) No ( ) Fairbank arrest test positive Yes ( ) No ( ) Positive J sign Yes ( ) No ( )
Foot— Angular Deformities	Hindfoot valgus Yes ( ) No ( ) Positive Jack test Yes ( ) No ( ) Positive tiptoe test Yes ( ) No ( ) Positive too-many-toes test Yes ( ) No ( )

**Table 3 medicina-59-01986-t003:** Distribution of musculoskeletal disorders (n = 22).

Musculoskeletal Disorders	n	%	CI 95%
Anisomelia	14	63.6	42.9–80.3
Hip dysplasia	1	4.5	0.8–21.8
Epiphysiolysis	1	4.5	0.8–21.8
Patellofemoral instability	13	59.1	38.7–76.7
Flatfoot valgus	21	95.4	78.2–99.2
Mechanical axis varus	5	22.7	10.1–43.4
Mechanical axis valgus	9	40.9	23.3–61.3

n: number of patients; %: percentage distribution; IC 95%: confidence interval 95%.

**Table 4 medicina-59-01986-t004:** Distribution of the number of musculoskeletal disorders per patient (n = 22).

Number of Musculoskeletal Disorders per Patient	Number of Patients	%	IC 95%
Patients with 1 musculoskeletal disorder	3	13.7	4.7–33.3
Patients with 2 musculoskeletal disorders	5	22.7	10.1–43.4
Patients with 3 musculoskeletal disorders	6	27.3	13.1–48.1
Patients with 4 musculoskeletal disorders	7	31.8	16.3–52.7
Patients with 5 musculoskeletal disorders	1	4.5	0.8–21.8

%: percentage distribution; IC 95%: confidence interval 95%.

**Table 5 medicina-59-01986-t005:** Indicators of the diagnostic capacity of physical examinations performed by two examiners.

Musculoskeletal Disorders	Examiner (E)	Sensitivity (%)	Specificity (%)	PPV (%)	NPV (%)	Accuracy (%)	AUC
Anisomelia	E1	85.71	33.33	85.71	33.33	76.47	0.595
	E2	85.71	66.67	92.31	50	82.35	0.761
Hip dysplasia	E1	0	65	0	92.86	62	0.325
	E2	0	80	0	94.2	76.19	0.40
Epiphysiolysis	E1	0	90	0	97.4	85.71	0.45
	E2	0	85	0	94.4	80.95	0.425
Flatfoot valgus	E1	90.48	100	100	33.33	90.91	0.952
	E2	90.48	100	100	33.33	90.91	0.952
Mechanical axis varus	E1	40	91.67	66.67	78.57	76.47	0.658
	E2	40	100	100	80	82.35	0.70
Mechanical axis valgus	E1	66.67	75	75	66.67	70.59	0.708
	E2	55.56	50	55.56	50	52.94	0.527

PPV: positive predictive value; NPV: negative predictive value; AUC: area under the ROC curve.

**Table 6 medicina-59-01986-t006:** Physical examination agreements between examiners 1 and 2.

Physical Examination		Kappa	CI 95%		*p*	Strength of Agreement *
Gait Evaluation	Claudication	0.49	0.42	0.94	0.007	Moderate
	Limb asymmetry (Galeazzi test)	0.20	0	0.65	0.158	Slight
	Trendelenburg sign	0.50	0.05	0.94	0.005	Moderate
	Lower limb asymmetry determined from measurement	0.32	0	0.74	0.056	Fair
Hip	Limitation on abduction	0.51	0	1.0	0.002	Moderate
	Hip flexion > 130°	- ^#^	- ^#^	- ^#^	- ^#^	- ^#^
	Extension > 30°	- ^#^	- ^#^	- ^#^	- ^#^	- ^#^
	Thomas test	0.11	0	0.72	0.284	Slight
	Hip internal rotation > 45°	0.65	0	1.0	0.0004	Substantial
	Hip external rotation > 50°	- ^#^	- ^#^	- ^#^	- ^#^	- ^#^
	Hip abduction > 45°	- ^#^	- ^#^	- ^#^	- ^#^	- ^#^
	Hip adduction > 40°	0.33	0	1.0	0.04	Fair
	Trendelenburg test	- ^#^	- ^#^	- ^#^	- ^#^	- ^#^
Knee	Varus axis	- ^#^	- ^#^	- ^#^	- ^#^	- ^#^
	Valgus axis	0.33	0	0.71	0.04	Fair
	Q-angle increase	0.31	0	0.70	0.06	Fair
	Position of the patella with the knee flexed: in front of the femoral condyles	- ^#^	- ^#^	- ^#^	- ^#^	- ^#^
	Positive patellar tilt test	0.07	0	0.47	0.364	Slight
	Fairbank arrest test positive	0.23	0	0.83	0.03	Fair
	Positive J sign	0.50	0	1.0	0.006	Moderate
Foot—Angular Deformities	Hindfoot valgus	0.70	0.30	1.0	0.0003	Substantial
	Positive Jack test	0.60	0.17	1.0	0.0007	Moderate
	Positive tiptoe test	0.22	0	0.62	0.118	Fair
	Positive too-many-toes test	0.50	0.05	0.94	<0.001	Moderate

* Strength of Agreement according to Landis and Koch (1977) [18]. ^#^ Kappa indicator could not be determined because one or more ratings were equal to 0.

## Data Availability

The data presented in this study are available on request from the corresponding author. The data are not publicly available due to ethical restrictions.

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
