# Peer review of "The Diagnostic Capacity of Physical Examinations in Diagnosing Musculoskeletal Disorders of the Lower Limbs in Children with Down Syndrome"

_medicina, 2023, doi:10.3390/medicina59111986_

Round 1
Reviewer 1 Report
Comments and Suggestions for Authors
Dear authors,
Your research is valuable in terms of its subject, but your manuscript has many major errors and looks sloppy. Unfortunately, I have to reject your inquiry. After the major corrections I will mention below, I recommend that you re-upload your research and evaluate it.
Abstract
The method section in the abstract is unnecessarily long and contains too much information. Please revise this section thoroughly. Additionally, information such as the number of patients that should be included in the method section is given in the findings. Revise the entire Abstract section from scratch.
Introduction
It contains a lot of general information in the introduction section and is written with only 5 references. There are no supporting studies to demonstrate the unique value of your research. After the 3rd paragraph, you tried to make your research original with only your own sentences. But what has been done in the literature? What's missing? and you need to thoroughly emphasize what you offer extra with this research. Research cannot go from the general to the specific purpose and hypothesis.
method
In the Method section, you should start with a good experimental design, explain the design of the study, and provide detailed patient information. You need to present detailed inclusion and exclusion criteria. I would also recommend that you include radiological images with explanations for readers to understand.
Discussion
You started the discussion with general information. Instead, start by giving the major findings of your research and discuss the research from many aspects on these findings with other references to get down to the conclusions that are important to you. I recommend that you detail the limitations of your research and write a reference-supported discussion rather than your own words.
best
Author Response
Dear reviewer,
We want to thank you for the pertinent comments and constructive criticisms. We have included the answers to your comments, point by point, below.
Modifications made according to the reviewers' suggestions are highlighted in blue in the revised version of the manuscript. We feel that our manuscript has improved after the revisions.
Thank you very much in advance for your kind attention,
Sincerely yours,
# 1) The method section in the abstract is unnecessarily long and contains too much information. Please revise this section thoroughly. Additionally, information such as the number of patients that should be included in the method section is given in the findings. Revise the entire Abstract section from scratch.
We revised the Abstract section with a profound restructuring of the methodology.
# 2) Introduction: It contains a lot of general information in the introduction section and is written with only 5 references. There are no supporting studies to demonstrate the unique value of your research. After the 3rd paragraph, you tried to make your research original with only your own sentences. But what has been done in the literature? What's missing? and you need to thoroughly emphasize what you offer extra with this research. Research cannot go from the general to the specific purpose and hypothesis.
We apologize for the error. Some statements were without their respective references. We recorded the references and, in addition, added two additional references.
# 3) In the Method section, you should start with a good experimental design, explain the design of the study, and provide detailed patient information. You need to present detailed inclusion and exclusion criteria. I would also recommend that you include radiological images with explanations for readers to understand.
We detail the study design in more depth at the beginning of the Methods section and more clearly present the inclusion and exclusion criteria. We added Figure 3 with some radiological diagnoses in the study.
# 4) Discussion: You started the discussion with general information. Instead, start by giving the major findings of your research and discuss the research from many aspects on these findings with other references to get down to the conclusions that are important to you. I recommend that you detail the limitations of your research and write a reference-supported discussion rather than your own words.
We restructured the beginning of the discussion section, starting with the main results of the study and subsequently presenting data from the literature. Some bibliographic references were added, and the study's limitations were highlighted.
Reviewer 2 Report
Comments and Suggestions for Authors
Thanks for the effort to complete such study
In the abstract you have to clarify who answered the questionnaire? the children or their parents .
Introduction :
According to data from the Brazilian Ministry of Health, a child is born with 35 DS for every 600 to 800 births , What about the total number of DS children in Brazil , to allow you to calculate sample size ?
22 child is small sample size – 4 patients had orthopedic problems . The sample size is not reasonable to reflect prvelance of MSD among DS . How did you calculate the sample size ?
Line 48:
devices, such as cardiovascular, remove devices into systems CVS , as devices may be confusing to the reader may be reflecting ( equipment- as pacemaker for example )
In methods :
physicians from different specialties , need to mention the specialties here
The study conducted in one year only DS pt’s were admitted to the clinic .
Did the patients answered questionnaire by their own ( online or paper based ) in the clinic or in their own time ? Did you consider educational level of guardians , or did they answer it during the appointment with an interpreter ?
Line112 patients could not have all of the proposed radiographs done due to technical difficulties (children's collaboration and acceptance . Did you exclude them from study result s ???
In Results :
Only four patients (18.2%) presented with orthopedic complaints , then you wrote that all 22 had msd??? This need clarification . 207Line
Assessing effectiveness of physical examination by only one pediatrician & orthopedist is not enough this should be considered limitation of the study . Also you didn’t mention the years of experience of physician 9 just mentioned that both has the same qualification and training sessions ) which is not enough ,May be if they were expert consultants , may be the results are different , so please ( consider years of experience of both examiners ) also much better if the pediatrician was orthopedist to have reasonable comparison between both examinations .
Discussion
Line 245
:change the word related to functional impairment into may cause
Line 247 remain upright posture
Line 251: However, regular follow-up of such patients 251 by orthopedists commonly does not follow protocols and depends on random referrals to 252 referenced services need reference ?
From ref (12 ) line 260 you mentioned that prevalence of 8% of this disorder in patients with DS ( need to mention their sample size and the setting of study )
Your findings : (1) 262 82.3% with Anisomelia; (2) 95.4% with flat foot valgus; (3) 52.9% with mechanical axis 263 valgus; and (4) 29.4% with mechanical axis varus. Limited reports have been published on the prevalence of the latter musculoskeletal disorders in children and adolescents with DS , What were their findings even if they were limited , please add their findings
] Line 266 : In our study, a single patient had epiphysiolysis. Children with endocrinopathies, mainly hypothyroidism , In your study did this single child had endocrinopathy?
Line276 : Furthermore, the agreement between the orthopedic physical examinations performed by the 277 two examiners was weak, poor, or indeterminate for most of the analyzed items , May be this is because of difference of specialty , this may be limitation of study also. Need to consider experience and practice of examiners not only the training protocol
Also you may add literature discussion accuracy of orthopedic physical examination of DS by different specialties ,
In REF list All Ref should be translated to English .
Comments on the Quality of English Language
Some minor modifications required regarding some terms as mentioned in the above section .Also all References should be in english
Author Response
Dear reviewer,
We want to thank you for the pertinent comments and constructive criticisms. We have included the answers to your comments, point by point, below.
Modifications made according to the reviewers' suggestions are highlighted in blue in the revised version of the manuscript. We feel that our manuscript has improved after the revisions.
Thank you very much in advance for your kind attention,
Sincerely yours,
# 1) In the abstract you have to clarify who answered the questionnaire? the children or their parents
We have corrected this excerpt in the summary: "Patients and guardians answered a simple questionnaire".
# 2) Introduction:
According to data from the Brazilian Ministry of Health, a child is born with 35 DS for every 600 to 800 births, What about the total number of DS children in Brazil, to allow you to calculate sample size? 22 child is small sample size – 4 patients had orthopedic problems. The sample size is not reasonable to reflect prevelance of MSD among DS. How did you calculate the sample size?
Unfortunately, the number of children with Down Syndrome in Brazil is not well established. It is estimated that 300 thousand people have Down Syndrome in Brazil, including children and adults. Therefore, we chose to estimate our study sample based on an expected sensitivity of 90% for diagnosing musculoskeletal disorders of the lower limbs following clinical examination, considering the prevalence of 70% of musculoskeletal alterations in patients with DS, with a 15% margin of error. In our study, all 22 patients were diagnosed with at least one musculoskeletal disorder.
The details on the sample calculation were corrected in the methods section.
Line 48: devices, such as cardiovascular, remove devices into systems CVS, as devices may be confusing to the reader may be reflecting( equipment- as pacemaker for example )
Thank you for your observation; the word device was removed from the text.
# 3) In Methods:
Physicians from different specialties, need to mention the specialties here by orthopedists and pediatricians
The term "Physicians from different specialties" was replaced by "orthopedists and pediatricians."
The study conducted in one year only DS pt’s were admitted to the clinic.
The study was developed at the genetic pediatrics outpatient clinic at the Botucatu Medical School Hospital, São Paulo State University (UNESP), Brazil. This information was added at the beginning of the Methods section.
Did the patients answered questionnaire by their own (online or paper based) in the clinic or in their own time ? Did you consider educational level of guardians, or did they answer it during the appointment with an interpreter?
Patients and guardians answered a simple questionnaire regarding orthopedic complaints applied by the same member of the research team responsible for asking the questions and recording the answers. This information was added at the Methods section (Clinical evaluations).
Line112 patients could not have all of the proposed radiographs done due to technical difficulties (children's collaboration and acceptance . Did you exclude them from study results???
Patients who did not undergo a specific radiograph were excluded from analyses related to these unperformed radiographs. All included patients underwent most of the proposed radiographs. This limitation was presented in the Discussion section.
# 4) In Results:
Only four patients (18.2%) presented with orthopedic complaints , then you wrote that all 22 had msd??? This need clarification . 207Line
Only four patients (18.2%) reported orthopedic complaints in their responses to the questionnaire. This information was clarified at the beginning of the results section.
Assessing effectiveness of physical examination by only one pediatrician & orthopedist is not enough this should be considered limitation of the study. Also you didn’t mention the years of experience of physician just mentioned that both has the same qualification and training sessions) which is not enough. Maybe if they were expert consultants, maybe the results are different, so please (consider years of experience of both examiners) also much better if the pediatrician was orthopedist to have reasonable comparison between both examinations.
The limitation related to the restricted number of examiners was added to the discussion section. Both physicians had the same experience time (5 years) and underwent qualification and training sessions before performing the systematized orthopedic examinations. This information was added to the methods section.
# 4) In Discussion:
Line 245 :change the word related to functional impairment into may cause
This correction was made on line 265.
Line 247 remain upright posture
This correction was made on line 267.
Line 251: However, regular follow-up of such patients by orthopedists commonly does not follow protocols and depends on random referrals to referenced services need reference ?
We added references to this excerpt on line number 273.
From ref (12 ) line 260 you mentioned that prevalence of 8% of this disorder in patients with DS ( need to mention their sample size and the setting of study )
We added this information to line 277. The reference related to this information has been corrected.
Your findings: (1) 262 82.3% with Anisomelia; (2) 95.4% with flat foot valgus; (3) 52.9% with mechanical axis 263 valgus; and (4) 29.4% with mechanical axis varus. Limited reports have been published on the prevalence of the latter musculoskeletal disorders in children and adolescents with DS, What were their findings even if they were limited , please add their findings.
This excerpt has been corrected, and a new reference has been added.
In our study, a single patient had epiphysiolysis. Children with endocrinopathies, mainly hypothyroidism, In your study did this single child had endocrinopathy?
The only patient diagnosed with epiphysiolysis was four years old, had congenital hypothyroidism, and was unable to walk since birth. This excerpt has been corrected in line 289.
Furthermore, the agreement between the orthopedic physical examinations performed by the two examiners was weak, poor, or indeterminate for most of the analyzed items. May be this is because of difference of specialty, this may be limitation of study also. Need to consider experience and practice of examiners not only the training protocol.
The limitation related to the restricted number of examiners, with little experience time and from different specialties, was presented as a study limitation in the discussion section.
Also you may add literature discussion accuracy of orthopedic physical examination of DS by different specialties
Unfortunately, we found no study in the literature that evaluated aspects of orthopedic physical examination in patients with Down Syndrome by different specialties. We believe that our study was the first to assess this aspect.
# 5) In REF list: All Ref should be translated to English
References have been reviewed and updated. We added English titles for all references.

Round 2
Reviewer 1 Report
Comments and Suggestions for Authors
Dear Authors,
The manuscript is ready to publish.
Best